# Mechanistic Insights into the Ameliorating Effect of Melanogenesis of Psoralen Derivatives in B16F10 Melanoma Cells

**DOI:** 10.3390/molecules27092613

**Published:** 2022-04-19

**Authors:** Yeji Lee, Chang-Gu Hyun

**Affiliations:** Jeju Inside Agency and Cosmetic Science Center, Department of Chemistry and Cosmetics, Jeju National University, Jeju 63243, Korea; cosmetics_chemistry@naver.com

**Keywords:** 5-hydroxypsoralen, 5-methoxypsoralen, 8-hydroxypsoralen, 8-methoxypsoralen, 5,8-dimethoxypsoralen, β-catenin, MAPK

## Abstract

The objectives of this study were to investigate the melanogenetic potential of the psoralen derivatives 5-hydroxypsoralen, 5-methoxypsoralen, 8-hydroxypsoralen, 8-methoxypsoralen, and 5,8-dimethoxypsoralen in B16F10 melanoma cells. The results indicated that melanin production is significantly stimulated in B16F10 melanoma cells with 5-methoxypsoralen, 8-methoxypsoralen, and 5,8-dimethoxypsoralen, especially for 5-methoxypsoralen (bergapten), as reported previously. In addition, Western blot results showed that the protein levels of microphthalmia-associated transcription factor (MITF), tyrosinase, tyrosinase-related protein-1 (TRP-1), and tyrosinase-related protein-2 (TRP-2) increase after bergapten treatment for the first time. The results also showed that bergapten promotes the phosphorylation of Akt at Ser 473 and glycogen synthase kinase-3β at Ser 9. Moreover, bergapten increased the content of β-catenin in the cell cytoplasm and nucleus by reducing the phosphorylated β-catenin (p-β-catenin) content. The results also indicated that bergapten regulates melanogenesis by upregulating the phosphorylation of p38 and JNK-mitogen-activated protein kinase. Taken together, these findings suggest that the regulation of melanogenesis by bergapten may be mediated by the β-catenin and MAPK signaling pathways and that bergapten might provide new insights into the pathogenesis of pigmented diseases.

## 1. Introduction

Melanin synthesis begins with the hydroxylation of l-tyrosine to l-DOPA and then to DOPA-quinone. These two reactions are catalyzed by the enzymatic activity of tyrosinase. Tyrosine and DOPA have long been known as substrates and positive modulators (hormone-like) of the melanogenesis pathway that determines metabolic fate [1,2]. Subsequently, DOPA-chrome is converted to 5,6-dihydroxy-indole-2-carboxylic acid (DHICA) in the presence of TRP-2/DOPA-chrome tautomerase (TRP-2/DCT). TRP-1/DHICA oxidase catalyzes the oxidation of DHICA to indole-5,6-quinone-2-carboxylic acid. These two closely related structures, TRP-2/DCT and TRP-1, act to form labile quinones, which undergo further polymerization to eventually produce brown-black eumelanin [3,4]. Therefore, to modulate the activity of tyrosinase, many different types of functional ingredients for modulating melanogenesis have been developed either synthetically or isolated from natural sources.

As shown in Figure 1a, psoralen is a linear, three-ring, heterocyclic, aromatic compound that possesses the furocoumarin entity, which consists of the coumarin-moiety-fused furan ring [5]. Psoralen is produced by the fruits, leaves, and seeds of *Angelica dahurica*, *Petroselinum crispum*, *Paliurus ramosissimus*, and *Peucedanum japonicum* and is found in all citrus fruits [6,7,8,9]. Psoralen is a natural product that has attracted much attention in the research community due to its wide range of biomedical applications in pharmaceuticals, ranging from photosensitizing, anti-inflammatory, insecticidal, and nematocidal to anti-microbial effects [10,11,12]. The most commonly used form of psoralen, in which it is combined with ultraviolet-A light, is for the treatment of psoriasis, vitiligo, and eczema [5]. This suggests that psoralen has broad prospects in therapeutic applications. Thus, several structural changes have been introduced to explore the role of specific positions with respect to the biological activity.

In our ongoing screening program to discover new cosmeceuticals and nutraceuticals, we have reported that several flavonoids and coumarins have anti-inflammatory, adipogenesis-inhibitory, and melanogenic activities [13,14,15,16,17]. As an extension of this study, we screened five psoralen derivatives—5-hydroxypsoralen (bergaptol), 5-methoxypsoralen (bergapten), 8-hydroxypsoralen (xanthotoxol), 8-methoxypsoralen (xanthotoxin), and 5,8-dimethoxypsoralen (isopimpinellin)—to identify the structural features involved in the melanogenesis of this class of molecules (Figure 1).

Several studies have been reported on the subject of our study, i.e., the functional effects of psoralen derivatives on the skin. Martins et al. [18] reported the development and validation of a physiologically based pharmacokinetic (PBPK) model of bergapten in various rat tissues, in which they attempted to infer human plasma and tissue concentrations of bergapten from the rat model. Matsuda et al. [19] measured the stimulation effect of melanogenesis by separating 16 coumarins from the seven Umbelliferae crude drugs. The results showed that xanthotoxin was the most powerful among the 16 coumarins at 100 μM concentrations, followed by isopimpinellin and bergapten, in terms of melanogenesis stimulation activity. Mengeaud et al. [20,21] confirmed the increase in tyrosinase, DOPA-chrome tautomerase (DCT), and tyrosinase-related protein-1 (TRP-1) protein production of bergapten extracted from *Citrus Bergamia* through Western blot experiments. Kinley et al. [22,23] reported quantitative data on epidermal melanogenesis by the ear and dorsal skin of pigmented, hairless rats exposed to sub-phototoxic UVA and simultaneously treated with psoralen derivatives, showing increased pigmentation in the order of 6,4,4′-trimethylangelicin > psoralen > 8-ethoxypsoralen > 5-methoxypsoralen > 4,4′,5′-trimethylazapsoralen. Recently, a Chinese research team designed and synthesized 4-methyl-6-phenyl-2*H*-furo [3,2-g]chromen-2-one (MPFC). They biologically evaluated the activity on melanogenesis and provided solid evidence showing that the p38 MAPK and PKA pathways are the targets of this compound for active melanin biosynthesis in murine B16 cells [24]. In addition, the application to vitiligo and the effect of increasing melanogenesis of psoralen derivatives have been studied, but the five psoralen derivatives in melanogenesis have not been compared and its mechanism of action is unknown, especially for 5-methoxypsoralen (bergapten). Therefore, this study focused on analyzing the effect of bergapten on the molecular mechanism of melanin biosynthesis in B16F10 mouse melanoma cells. We reported that the melanogenic effects of bergapten depend on increased MITF gene expression, which is mediated by the activation of both the β-catenin and MAPK signal pathways.

## 2. Results

### 2.1. Effect of Psoralen Derivatives on the Viability of B16F10 Cells

To investigate whether psoralen derivatives exert cytotoxicity on B16F10 cells, the cells were treated with various concentrations (12.5, 25, 50, 100, 200, and 400 μM) of psoralen derivatives and α-MSH (100 nM) for 72 h. There were no significant differences for up to 25 μM of psoralen derivatives in B16F10 cells (Figure 2). Therefore, we used psoralen derivatives with concentrations below 25 μM for further experiments.

### 2.2. Effect of Psoralen Derivatives on the Melanin Content of B16F10 Cells

To examine whether psoralen derivatives affect the melanogenesis of B16F10 cells, the cells were treated with psoralen derivatives (6.25, 12.5, and 25 μM) and α-MSH (100 nM) for 72 h. α-MSH was used as a positive control. Bergapten, xanthotoxin, and isopimpinellin increased melanin content compared to the untreated control group. Among them, the melanin content enhanced the effect of bergapten more than other psoralen derivatives (Figure 3). Therefore, further experiments were performed to evaluate the melanogenesis effects of bergapten.

### 2.3. Effect of Psoralen Derivatives on Tyrosinase Activity in B16F10 Cells

The effects of bergapten on tyrosinase activity in B16F10 cells are shown in Figure 4. The cellular tyrosinase activity was significantly increased by bergapten compared to the untreated control group. When compared with the untreated control, bergapten increased tyrosinase activity to 223.7% at 25 μM. These results are consistent with those that compared the effects of bergapten on the melanin content of B16F10 cells.

### 2.4. Effect of Bergapten on the Expression of Melanogenic Enzymes and MITF

To determine the effect of bergapten on the expression of melanogenic enzymes (tyrosinase, TRP-1, and TRP-2) and MITF, Western blot experiment was conducted. As shown in Figure 5, the protein expression of tyrosinase, TRP-1, and TRP-2 was significantly increased by bergapten compared to the untreated control group. Moreover, the MITF expression was also increased by bergapten. These results show that bergapten enhances melanogenesis enzymes by increasing MITF expression and increases melanogenesis.

### 2.5. Effect of Bergapten on the Wnt/β-Catenin Signaling Pathway

It has been reported that GSK3β (Ser 9) phosphorylated in the Wnt/β-catenin pathway induces the accumulation of β-catenin in the cytoplasm and that accumulated β-catenin is transported to the nucleus to increase MITF expression. We investigated whether bergapten induces melanogenesis through the Wnt/β-catenin signal in B16F10 cells. The results showed that bergapten increases P-GSK3β (Ser 9) and β-catenin compared to the untreated group. However, bergapten inhibited P-β-catenin expression compared to the untreated group. These results suggest that bergapten increases melanogenesis through the Wnt/β-catenin signaling pathway (Figure 6).

### 2.6. Effect of Bergapten on the AKT Signaling Pathway

It has been reported that phosphorylated AKT in the AKT signaling pathway inhibits melanin synthesis. That is, the inhibition of AKT phosphorylation induces melanin production. Therefore, we investigated whether bergapten induces melanogenesis through the AKT signaling pathway in B16F10 cells. The results showed that bergapten decreases the phosphorylation of AKT. These results indicate that bergapten induces melanogenesis by reducing the phosphorylation of AKT in the AKT signaling pathway (Figure 7).

### 2.7. Effect of Bergapten on the PKA Signaling Pathway

We examined whether bergapten induces melanogenesis through the PKA signal in B16F10 cells. As shown in Figure 8, bergapten increased PKA phosphorylation compared to the untreated group. These results suggest that bergapten increases melanogenesis through the PKA signaling pathway.

### 2.8. Effect of Bergapten on the MAPK Signaling Pathway

We investigated whether bergapten affects MAPK phosphorylation in B16F10 cells. The results showed that bergapten increases the phosphorylation of ERK, JNK, and p38 in B16F10 cells. These results indicate that bergapten induces melanogenesis by reducing the phosphorylation of JNK and p38 in the MAPK signaling pathway (Figure 9).

## 3. Discussion

Drug repurposing or natural product repurposing, as an emerging alternative strategy based on the assumption that a single compound can target multiple targets, has demonstrated the applicability of multi-target drugs and polypharmacology across multiple therapeutic areas in several studies during the 21st century [25]. Psoralene and its derivatives belong to an important group of natural compounds found in vegetables and plants. They have been extensively studied for the phototreatment of psoriasis, vitiligo, atopic dermatitis, and other dermatological pathologies, as well as cutaneous T-cell lymphoma, and are considered representative multi-targeting agents [26,27]. Therefore, these small molecules still represent an attractive scaffold for further development and application in several therapeutic fields.

From the use of our screening strategy for multi-targeting agents, our laboratory reported attractive results that elucidate the functionalities of various flavonoids and coumarins in terms of their anti-inflammatory, anti-obesity, and melanogenesis-inhibiting/-stimulating effects [13,14,15,16,17,28,29,30,31]. In this study, we tried to verify the efficacy of psoralene and systematically applied it to skin-related diseases using five psoralen derivatives: 5 hydroxypsoralen, 5-methoxypsoralen, 8-hydroxypsoralen, 8-methoxypsoralen, and 5,8-dimethoxypsoralen (Figure 3). In the present study, we showed the inhibitory effects of psoralen derivatives on melanogenesis based on their capacity to inhibit melanin content and cellular tyrosinase activity. In addition, 5-methoxypsoralen (bergapten) exhibited more stimulatory effects of melanin production in mouse B16F10 melanoma cells than other psoralene derivatives (Figure 4). In terms of the structure–activity relationship, it may be assumed that a simple structure of psoralen with a methoxy group each at positions 5 and 8, such as xanthotoxin, bergapten, and isopimpinellin, is preferable for stimulatory activity. The introduction of hydroxyl groups into the psoralen furocoumarin ring at positions 5 and 8 was not advantageous. In addition, the interesting fact is that isopimpinellin, which has methoxy groups at both positions 5 and 8, is disadvantageous compared to xanthotoxin and bergapten, in which only one methoxy group was introduced, at either position 5 or position 8 of the ring.

This study showed that bergapten activates melanogenesis by increasing the protein levels of tyrosinase, tyrosinase-related protein-1 (TRP-1), and tyrosinase-related protein-2 (TRP-2) in B16F10 cells, suggesting that bergapten activates melanin production and intracellular tyrosinase activity by increasing the expression of the melanogenic enzymes (Figure 5). These results are consistent with those of previous studies, which have reported that 5-methoxypsoralen has an melanogenesis-stimulating effect [20,21]. However, the underlying molecular mechanisms remain to be explored.

Since microphthalmia-associated transcription factor (MITF) plays a pivotal role in melanogenesis as the major transcriptional regulator of melanogenic proteins such as tyrosinase [32,33], TRP-1, and TRP-2, we demonstrated that the expression of MITF increased by bergapten (6.25, 12.5, and 25 μM) for 48 h. The activation of MITF, a transcription factor that regulates tyrosinase gene expression, is known to be a critical event in melanogenesis. Thus, it was suggested that one of the mechanisms for bergapten-induced melanogenesis in B16F10 cells is the increase in melanogenic protein expression through the positive regulator MITF.

There are some melanogenesis-related signal pathways, such as the Wnt/β-catenin signal pathway, the PI3K/Akt signal pathway, the PKA signal pathway, and the mitogen-activated protein kinase (MAPK) signal pathway, that are certainly involved in this process so as to affect the melanogenesis-related transcription factor MITF [32].

The GSK-3β and β-catenin proteins are composed of the Wnt/β-catenin signal pathway. GSK-3β is a constitutive active kinase and is phosphorylated by several kinases, including Akt and PKA. Recent studies have shown that the β-catenin signaling pathway is closely related to melanin synthesis since it promotes MITF transcription after β-catenin transfer from the cytoplasm to the nucleus, thereby upregulating the expression of MITF and then binding with lymphocyte enhancing factor (LEF) [33,34]. In this study, we found that bergapten promotes the phosphorylation of GSK-3β, which leads to the accumulation of β-catenin in the cytoplasm. Consistent with this research, our results showed that bergapten promotes the accumulation of β-catenin in the cytoplasm, which leads to the overexpression of MITF and eventually promotes the biosynthesis of melanin (Figure 6). Thus, melanin production mediated by compound bergapten was probably triggered through Wnt pathways, consistent with reports revealing that the activation of GSK3/β-catenin influenced melanin production in B16F10 melanoma cells.

In contrast, PI3K/Akt pathways can phosphorylate MITF to regulate post-transcriptional activity. However, the activation of the PI3K/Akt pathway in melanogenesis is controversial. Kim et al. [35] reported that C2-ceramide-mediated depigmentation in Mel-Ab cells occurs through a decrease in p-Akt levels. In contrast, there is also evidence showing that elevated p-Akt levels inhibit melanin synthesis [36,37]. Our data showed that p-Akt levels are downregulated in bergapten-induced melanogenesis in B16F10 cells (Figure 7). This suggests that the PI3K/Akt pathway may contribute to the stimulation of melanogenesis by bergapten.

It is well known that the PKA signaling pathway is also involved in melanogenesis. PKA can be activated by the elevation of cellular cAMP and increases MITF transcriptional activity through CREB phosphorylation, resulting in the protein expression of tyrosinase, TRP-1, and TRP-2 [38,39]. Therefore, to determine whether the effect of bergapten on melanin synthesis is mediated by the PKA signaling pathway, Western blot experiments were conducted to determine whether PKA phosphorylation was enhanced. As shown in Figure 8, melanin synthesis induced by bergapten strongly activated PKA phosphorylation in a concentration-dependent manner, suggesting that the PKA signaling pathway is involved in bergapten-induced melanin synthesis.

Furthermore, it has been shown that the MAPK signaling pathway participates in the activation of MITF in the process of melanogenesis. In addition, the activation of JNK and p38 MAPK and the inhibition of the ERK signaling have been reported to be related to the stimulation of pigmentation by increasing the tyrosinase activity [40,41,42,43]. Therefore, we investigated whether bergapten induces JNK and p38 MAPK or ERK repression in B16F10 cells. Figure 9 shows that bergapten treatment led to a significant increase in the phosphorylation of p38 MAPK at 4 h, whereas no effect was observed in the levels of the phosphorylation of ERK (Figure 9). These results indicate that bergapten may enhance the phosphorylation of JNK and p38 MAPK, leading to melanogenesis in B16F10 cells. This result suggests that JNK and p38 MAPK functionally regulated bergapten-induced melanin formation and the induction of tyrosinase and MITF expression in B16F10 cells.

Taken together, our results suggest that activating MITF through the Akt/GSK3β/β-catenin, PKA, and MAPK pathways mediates the hypopigmentary effect of bergapten. However, our study had several limitations. The mouse melanoma cancer cell B16F10 is widely used as an experimental cell line in many studies on melanogenesis. Therefore, we used the same cell line for experiments in this study. Although bergapten showed the melanogenic effect at the cellular level, these results do not always provide the same outcomes as human melanocytes or clinical manner [44]. Therefore, for practical application of the results, further human melanocytes or clinical studies will be required to determine the therapeutic regimen of bergapten for the treatment of hypopigmentary disorders in humans.

## 4. Materials and Methods

### 4.1. Chemicals and Reagents

8-hydroxypsoralen, 5-hydroxypsoralen, 5-methoxypsoralen, and 5,8-dimethoxypsoralen were purchased from ChemFaces (Wuhan, China) and 8-methoxypsoralen was purchased from TCI (Tokyo, Japan). Protease inhibitor cocktail, α-melanocyte-stimulating hormone (α-MSH), sodium hydroxide (NaOH), and l-DOPA were purchased from Sigma-Aldrich (St. Louis, MO, USA); and 3-(4,5-dimethylthiazol-2-yl)-2,5-diphenyltetrazolium bromide (MTT) and dimethyl sulfoxide (DMSO) were purchased from Biosesang (Seongnam, Gyeonggi-do, Korea). Dulbecco’s Modified Eagle’s Medium (DMEM) and penicillin-streptomycin were purchased from Thermo Fisher Scientific (Waltham, MA, USA). Fetal bovine serum (FBS) was purchased from Merck Millipore (Burlington, MA, USA). The primary antibodies used for Western blot were p-ERK, ERK, p-JNK, JNK, p-p38, p38, p-AKT, AKT, p-GSK-3β, GSK-3β, p-β-catenin, β-catenin, p-PKA, PKA, and β-actin. They were purchased from Cell Signaling Technology (Danvers, MA, USA). All reagents used were of analytical grade.

### 4.2. Cell Culture

B16F10 mouse melanoma cells were purchased from the ATCC (Manassas, VA, USA). The cells were cultured in DMEM supplemented with 1% penicillin/streptomycin and 10% FBS at 37 °C under a humidified incubator (NB-203XL, N-BIOTEK, Inc., Bucheon, South Korea) of 5% CO_2_. B16F10 cells were subcultured every 4 days. The experiments were implemented when the cells showed a 90% confluence density [45,46].

### 4.3. Cell Viability

Cell viability was measured using the MTT assay. B16F10 cells were seeded in 24-well plate at 8.0 × 10^3^ cells/well and incubated for 24 h. Then, the cells were treated with various concentrations of samples (12.5, 25, 50, 100, 200, and 400 μM) for 72 h. MTT reagent (0.4 mg/mL) was treated with the medium for 3 h. The medium was removed, DMSO was added into each well to dissolve purple formazan crystals, and absorbance was measured at 570 nm using a spectrophotometric microplate reader (Epoch, BioTek Instruments, Winooski, VT, USA)

### 4.4. Melanin Content

l-tyrosine and l-DOPA, two major substrates of the melanogenesis pathway, show a positive regulation of melanin pigmentation. Many studies have reported using B16 mouse melanoma cells cultured in DMEM characterizing a high concentration of tyrosine (0.4 mM), which can affect cellular phenotype and melanogenesis [47,48,49]. Therefore, we conducted all experiments within 84 h of the start of the log phase of B16F10 cells in order to minimize the unexpected role of DMEM medium. B16F10 cells were seeded in a 60 mm cell culture dish at 8.0 × 10^4^ cells/dish for 24 h. The cells were exposed to various concentrations of samples (6.25, 12.5, and 25 μM), and α-MSH (100 nM) was used for treatment for 72 h. After incubation, the cells were washed with 1 × PBS buffer and lysed using lysis buffer (RIPA buffer; 1% protease inhibitor cocktail) at 4 °C for 10 min. Then, the cells were scraped with a cell scraper, and lysates were vortexed three times at 5 min intervals. After centrifugation for 20 min at 15,000 rpm and at −8 °C, the supernatant was removed to obtain pellets. The cell pellets were dissolved in 1 N NaOH supplemented with 10% DMSO at 80 °C for 10 min. Cell lysates were transferred to a 96-well plate and measured at 405 nm using a microplate reader (Epoch, BioTek Instruments, Winooski, VT, USA).

### 4.5. Tyrosinase Activity

B16F10 cells were seeded in a 60 mm cell culture dish at 8.0 × 10^4^ cells/dish for 24 h. Cells were treated with increasing doses of samples and α-MSH (100 nM) for 72 h. After incubation, the cells were washed with 1 × PBS buffer and lysed with lysis buffer (RIPA buffer; 1% protease inhibitor cocktail) at 4 °C for 10 min. Then, the cells were scraped with a cell scraper. Lysates were centrifuged at 15,000 rpm at −8 °C for 20 min to obtain supernatants. The protein level was quantified using a BCA protein assay kit. Then, 20 μL of an adjusted protein sample and 80 μL of l-DOPA (final 8 mM) were added to a 96-well plate to measure tyrosinase activity. Corrections for auto-oxidation of l-DOPA in controls were made. After incubation at 37 °C for 1 h, absorbance was measured at 490 nm using a microplate reader (Epoch, BioTek Instruments, Winooski, VT, USA).

### 4.6. Western Blotting

B16F10 cells were seeded in a 60 mm cell culture dish at 8.0 × 10^4^ cells for 24 h. The cells were treated with 5-methoxypsoralen (6.25, 12.5, and 25 μM) for each protein expression time. After incubation, they were washed with 1 × PBS buffer and lysed using a lysis buffer at 4 °C for 10 min. Then, the cells were scraped with a cell scraper and transferred to a 1.5 mL e-tube. After vortexing three times at 10 min intervals, lysates were centrifuged at 15,000 rpm at −8 °C for 20 min to obtain a supernatant. The protein level was quantified using a BCA protein assay kit, and the sample was heated at 100 °C for 5 min. Twenty micrograms of proteins of each sample was loaded on 10% (*v*/*v*) sodium dodecyl sulfate-polyacrylamide gel (SDS-PAGE). The proteins were separated by size by electrophoresis. Proteins were transferred to a polyvinylidene fluoride (PVDF) membrane. The membrane was blocked with 5% (*w*/*v*) skim milk for 1 h and washed with 0.1% Tween 20 (TBS-T) for 10 min a total of six times. The membrane was incubated overnight at 4 °C with the primary antibody (1:1000). Then, the membranes were washed with 1 × TBS-T and reacted for 2 h at room temperature using a secondary antibody (1:2000). After washing, specific proteins were detected using an ECL kit.

### 4.7. Statistical Analyses

All experiment results were expressed as the mean ± standard deviation (SD) of at least three independent experiments. Statistical analyses were performed using Student’s t-tests or one-way ANOVA using IBM SPSS (v. 20, SPSS Inc., Armonk, NY, USA). *p*-Values < 0.05 (*) or 0.01 (**) were marked as statistically significant.

## Figures and Tables

**Figure 1 molecules-27-02613-f001:**
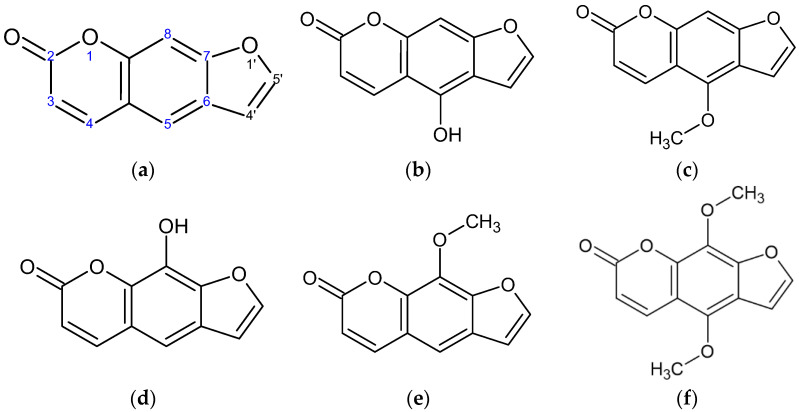
Structure of psoralen derivatives. (**a**) Psoralen, (**b**) 5-hydroxypsoralen (bergaptol), (**c**) 5-methoxypsoralen (bergapten), (**d**) 8-hydroxypsoralen (xanthotoxol), (**e**) 8-methoxypsoralen (xanthotoxin), and (**f**) 5,8-dimethoxypsoralen (isopimpinellin).

**Figure 2 molecules-27-02613-f002:**
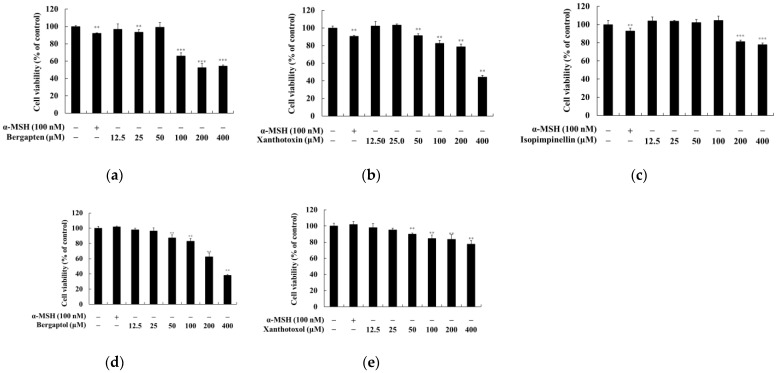
Effects of psoralen derivatives on cell viability in B16F10 melanoma cells. The cells were treated with psoralen derivatives (12.5, 25, 50, 100, 200, and 400 μM) and α-MSH (100 nM) for 72 h. Cell viability of B16F10 cells subjected to (**a**) 5-methoxypsoralen (bergapten), (**b**) 8-methoxypsoralen (xanthotoxin), (**c**) 5,8-dimethoxypsoralen (isopimpinellin), (**d**) 5-hydroxypsoralen (bergaptol), and (**e**) 8-hydroxypsoralen (xanthotoxol) was measured by MTT assay. The results are presented as the mean ± SD of three independent experiments. *** *p* < 0.001, ** *p* < 0.01 vs. the untreated group.

**Figure 3 molecules-27-02613-f003:**
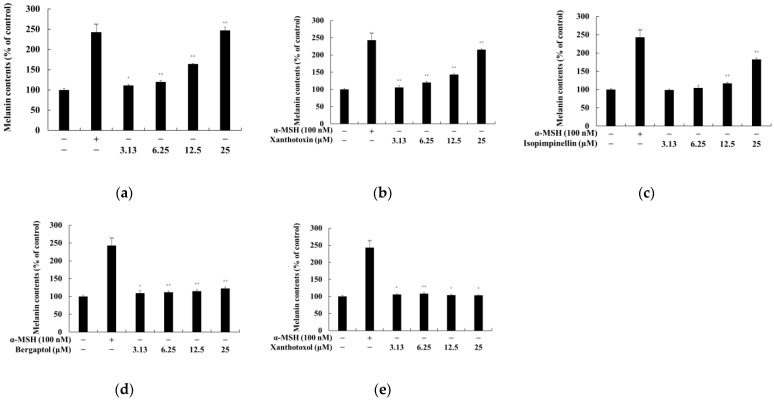
Effects of psoralen derivatives on the melanin content in B16F10 melanoma cells. The cells were treated with psoralen derivatives (3.13, 6.25, 12.5, and 25 μM) and α-MSH (100 nM) for 72 h. Melanin content of B16F10 cells subjected to (**a**) 5-methoxypsoralen (bergapten), (**b**) 8-methoxypsoralen (xanthotoxin), (**c**) 5,8-methoxypsoralen (isopimpinellin), (**d**) 5-hydroxypsoralen (bergaptol), and (**e**) 8-hydroxypsoralen (xanthotoxol). The results are presented as the mean ± SD of three independent experiments. * *p* < 0.05; ** *p* < 0.01 vs. the untreated group.

**Figure 4 molecules-27-02613-f004:**
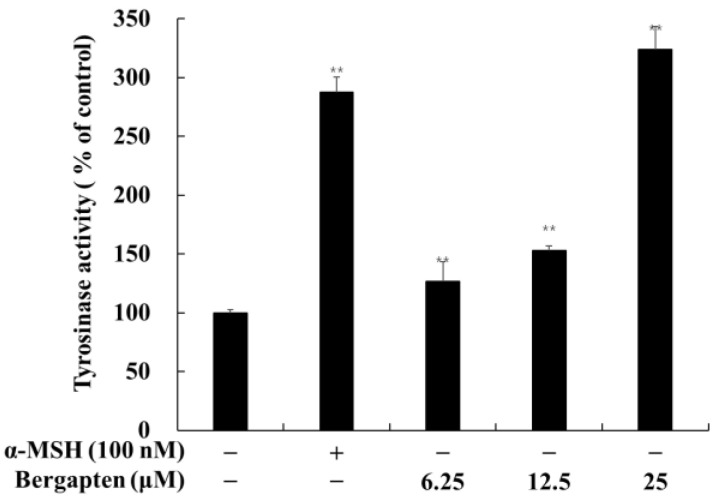
Effects of bergapten on tyrosinase activity in B16F10 melanoma cells. The cells were treated with bergapten (6.25, 12.5, and 25 μM) and α-MSH (100 nM) for 72 h. α-MSH was used as a positive control. The results are presented as the mean ± SD of three independent experiments. ** *p* < 0.01 vs. the untreated group.

**Figure 5 molecules-27-02613-f005:**
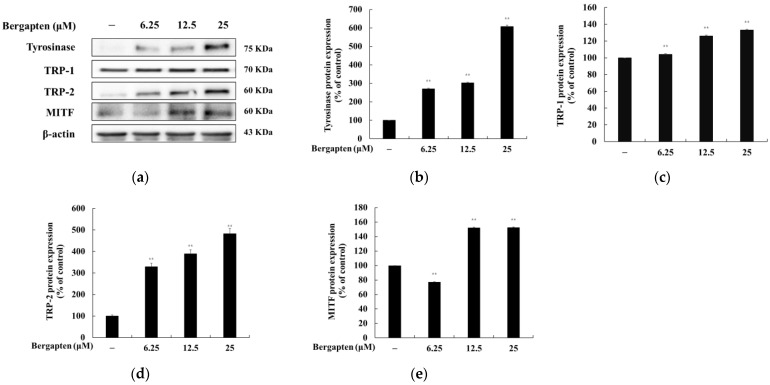
Effects of bergapten on the protein expression level of tyrosinase, TRP-1, TRP-2, and MITF in B16F10 melanoma cells. The cells were treated with bergapten (6.25, 12.5, and 25 μM) for 48 h. (**a**) Western blotting results and protein expression of (**b**) tyrosinase, (**c**) TRP-1, (**d**) TRP-2, and (**e**) MITF. β-actin was used as a loading control. The results are presented as the mean ± SD of three independent measurements using Image J. ** *p* < 0.01 vs. untreated group.

**Figure 6 molecules-27-02613-f006:**
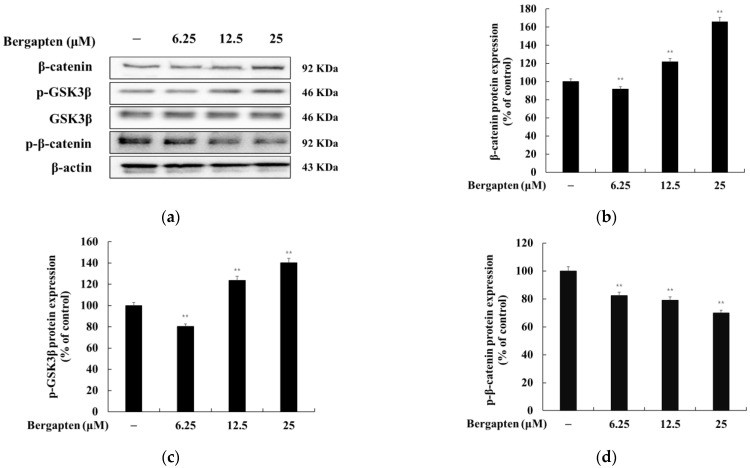
Effects of bergapten on the protein expression level of β-catenin, p-GSK3β, and p-β-catenin in B16F10 melanoma cells. The cells were treated with bergapten (6.25, 12.5, and 25 μM) for 24 h. (**a**) Western blotting results and protein expression of (**b**) β-catenin, (**c**) P-GSK3β/T-GSK3β, and (**d**) P-β-catenin. β-actin was used as a loading control. The results are presented as the mean ± SD of three independent measurements using Image J. ** *p* < 0.01 vs. the untreated group.

**Figure 7 molecules-27-02613-f007:**
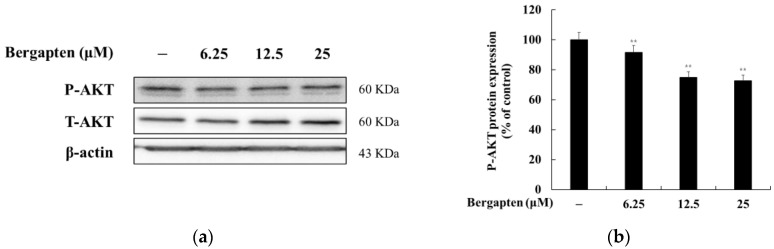
Effects of bergapten on the phosphorylation level of AKT in B16F10 melanoma cells. The cells were treated with bergapten (6.25, 12.5, and 25 μM) for 4 h. (**a**) Western blotting results and protein expression of (**b**) P-AKT/T-AKT. β-actin was used as a loading control. The results are presented as the mean ± SD of three independent measurements using Image J. ** *p* < 0.01 vs. the untreated group.

**Figure 8 molecules-27-02613-f008:**
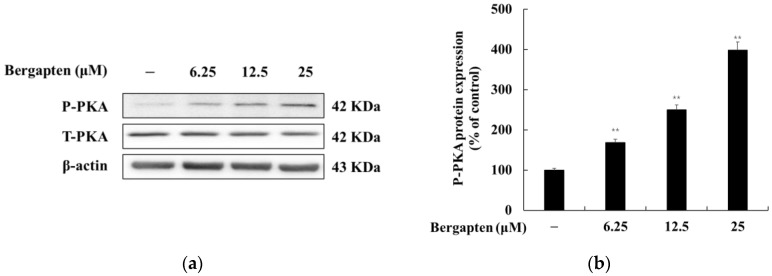
Effects of bergapten on the phosphorylation level of PKA in B16F10 melanoma cells. The cells were treated with bergapten (6.25, 12.5, and 25 μM) for 20 h. (**a**) Western blotting results and protein expression of (**b**) P-PKA/T-PKA. β-actin was used as a loading control. The results are presented as the mean ± SD of three independent measurements using Image J. ** *p* < 0.01 vs. the untreated group.

**Figure 9 molecules-27-02613-f009:**
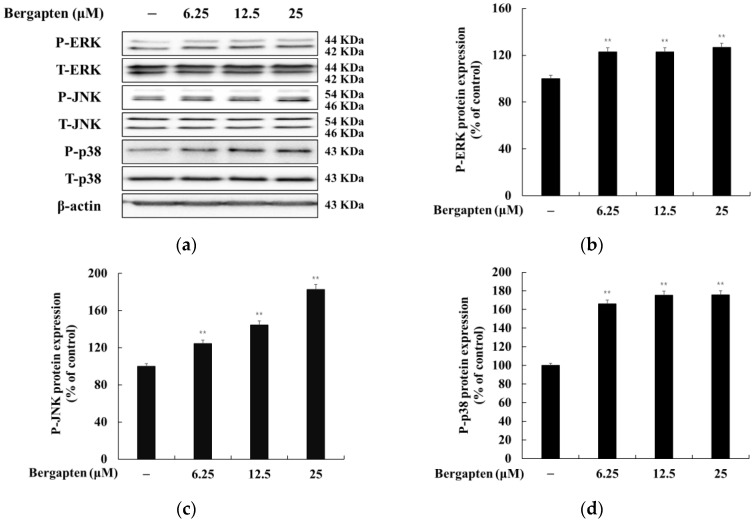
Effects of bergapten on the phosphorylation level of MAPKs in B16F10 melanoma cells. The cells were treated with bergapten (6.25, 12.5, and 25 μM) for 4 h. (**a**) Western blotting results and protein expression of (**b**) P-ERK/T-ERK, (**c**) P-JNK/T-JNK, and (**d**) P-p38/T-p38. β-actin was used as a loading control. The results are presented as the mean ± SD of three independent measurements using Image J. ** *p* < 0.01 vs. the untreated group.

## Data Availability

Not applicable.

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
