# Peer review of "Mechanistic Insights into the Ameliorating Effect of Melanogenesis of Psoralen Derivatives in B16F10 Melanoma Cells"

_molecules, 2022, doi:10.3390/molecules27092613_

Round 1

Reviewer 1 Report

The manuscript "Mechanistic insights into the ameliorating effect of melanogenesis of psoralen derivatives in B16F10 melanoma cells" provides information on the antimelanogenic activity of psoralen analogs. However, there are previous works on the antimelanogenic activity of psoralen, such as Yin et al., "A novel psoralen derivative-MPFC enhances melanogenesis via activation of p38 MAPK and PKA signaling pathways in B16 cells". In order for the work to be considered relevant, the authors must add positive controls to the biological assays and carry out the respective comparisons in terms of potency and efficacy of these analogs.

On the other hand, the authors should add a discussion about the structure-activity relationship of these analogs, indicating how each of the substituents or residues of these analogs influence the activity.

Finally, the authors must indicate what was the criterion to establish the number of cells per well, if it is necessary to put a reference.

Author Response

Thank you for your useful comments and suggestions on the language and structure of our manuscript. We have modified the manuscript accordingly, and detailed corrections are listed below point by point. Our authors received English proofreading through native speakers before submitting paper, but we will be making corrections in English for the newly changed and inserted content.

[Reviewer 1]

(1) The manuscript "Mechanistic insights into the ameliorating effect of melanogenesis of psoralen derivatives in B16F10 melanoma cells" provides information on the antimelanogenic activity of psoralen analogs. However, there are previous works on the antimelanogenic activity of psoralen, such as Yin et al., "A novel psoralen derivative-MPFC enhances melanogenesis via activation of p38 MAPK and PKA signaling pathways in B16 cells". In order for the work to be considered relevant, the authors must add positive controls to the biological assays and carry out the respective comparisons in terms of potency and efficacy of these analogs.

→ As pointed out by the reviewer, we inserted the valuable research results of Yin et al (2018) in the introduction section. (L73-77). Regarding the positive control, α-MSH (100 nM) serves as a positive control, and the following four psoralen derivatives (xanthotoxin, isopimpinellin, bergaptol, xanthotoxol) except 5-methoxypsoralen (bergapten) serve as controls.

(2) On the other hand, the authors should add a discussion about the structure-activity relationship of these analogs, indicating how each of the substituents or residues of these analogs influence the activity.

→ We modified the association between structure-activity and melanogenesis in more depth. (L215-222)

(3) Finally, the authors must indicate what was the criterion to establish the number of cells per well, if it is necessary to put a reference.

→ In general, experiments are conducted based on a state of 80-90% or more confluence. Prior to setting up the experimental model, seeding the same level of cells and pre-culturing for at least 16-18 hr including 24 hr for cell stabilization until cell attachment. We have inserted relevant content and references (44-45) in Materials and methods. (L315-316)

Reviewer 2 Report

The topic is of interest. However, there are several weaknesses that have to be corrected.

Some of them may originate in a limited expertise of the authors in melanin pigmentation as documented by random and not always proper use of references and some methodological nuances. I suggest some reading about hormonal regulation of melanogenesis (Physiol Rev 84, 1155-1228, 2004) and the role of melanin precursors in this process (Pigment Cell Melanoma Res  25, 14-27, 2012).

In general, B16 melanoma is a faulty model to study regulation of human pigmentation. It is a murine cell line that is malignant and this has to be acknowledged in limitations. Ideally some crucial experiments could be repeated on normal human melanocytes to validate the conclusions as relates to physiological regulation of melanin pigmentation.

The concentrations of test compounds are high (10-5 M range), which should be acknowledged.

Clarifications concerning tyrosinase assays are required. 

For example, ideally dopa oxidase activity of tyrosinase should be performed using 1 mM L-DOPA at 475 nm. This would allow to present data in nmols of dopachrome of molar coefficient for 475 nm wavelength is defined. It is unclear whether the absorbancy due to DOPA auto-oxidation in the buffer was subtracted. Ideally, negative control should be represented by heat inactivated extract, since Cu can oxidate DOPA.

Why tyrosine hydroxylase activity of tyrosinase was not measured?

Authors used DMEM high in tyrosine concentration, probably not even realizing that tyrosine can stimulate melanogenesis in melanoma cells (J Cell Sci 89, 287-296, 1988), which is relevant to this studies.

I would show the entire WB as supplemental file, they are very nice and clean.

Mechanism of action could be discussed in more educated manner. See recommended reading above. Also what is a target regulatory protein for psolaren derivatives?

Author Response

Thank you for your useful comments and suggestions on the language and structure of our manuscript. We have modified the manuscript accordingly, and detailed corrections are listed below point by point. Our authors received English proofreading through native speakers before submitting paper, but we will be making corrections in English for the newly changed and inserted content.

[Reviewer 2]

(1) The topic is of interest. However, there are several weaknesses that have to be corrected. Some of them may originate in a limited expertise of the authors in melanin pigmentation as documented by random and not always proper use of references and some methodological nuances. I suggest some reading about hormonal regulation of melanogenesis (Physiol Rev 84, 1155-1228, 2004) and the role of melanin precursors in this process (Pigment Cell Melanoma Res 25, 14-27, 2012).

→ We fully agree with the reviewer's point of reference for the paper and the need for an introduction to the scientific story. We are well aware that scientists and publishers' excessive attention to the impact factor (IF) is contributing to the lack of bibliography. Ultimately, we revised the reviewer-recommended article and related content in the introduction section. (L27-38)

(2) In general, B16 melanoma is a faulty model to study regulation of human pigmentation. It is a murine cell line that is malignant and this has to be acknowledged in limitations. Ideally some crucial experiments could be repeated on normal human melanocytes to validate the conclusions as relates to physiological regulation of melanin pigmentation.

→ Although many papers studying melanogenesis using melanoma cells are still published, we admit that normal melanocytes are a more accurate model, as the reviewers point out. Although it was not possible to revise all the study contents with human melanocytes within the 10-day revision period, the melanin content were checked using human epidermal melanocytes. Unfortunately, although no clean data were obtained, we have also identified a tendency for bergapten to increase melanin content in Human Epidermal Melanocytes (see Appendix).

(3) The concentrations of test compounds are high (10-5 M range), which should be acknowledged.

→ The highest concentration we applied, 25 μM, is not considered a high concentration as it corresponds to 0.54 mg when applied to a 100 cc topical product. In fact, we are submitting research results related to inflammation in another paper, and among the data presented in this paper, human skin irritation test did not find any skin irritation. (see appendix).

(4) Clarifications concerning tyrosinase assays are required. For example, ideally dopa oxidase activity of tyrosinase should be performed using 1 mM L-DOPA at 475 nm. This would allow to present data in nmols of dopachrome of molar coefficient for 475 nm wavelength is defined. It is unclear whether the absorbancy due to DOPA auto-oxidation in the buffer was subtracted. Ideally, negative control should be represented by heat inactivated extract, since Cu can oxidate DOPA.

→ We apologize for the typo in the experimental method of our paper. Absorbance was measured at 490 nm not 450 nm. (L345)

(5) Why tyrosine hydroxylase activity of tyrosinase was not measured?

→ There is no particular reason for not measuring the tyrosine hydroxylase activity of tyrosinase. Since most melanogenesis-related papers use only one substrate, either tyrosine or dopa, we also referred to it.

(6) Authors used DMEM high in tyrosine concentration, probably not even realizing that tyrosine can stimulate melanogenesis in melanoma cells (J Cell Sci 89, 287-296, 1988), which is relevant to this studies.

→ Thank you for introducing this interesting paper. According to this paper, although dopa, a substrate, affects melanogenesis, there is no problem because 5-hydroxypsoralen (bergaptol) and 8-hydroxypsoralen (xanthotoxol) act as negative controls. Rather, we were concerned about DMSO as a solvent for psoralen derivatives. So, we studied the effect of DMSO with melanogenesis. As shown in the appendix, DMSO did not affect melanogenesis up to a high concentration of 1%.

(7) I would show the entire WB as supplemental file, they are very nice and clean.

(8) Mechanism of action could be discussed in more educated manner. See recommended reading above. Also what is a target regulatory protein for psolaren derivatives?

→ In relation to the discussion of educational content pointed out by reviewers, we additionally prepared a schematic diagram of the target protein and mechanism as Figure 10, considering that it is a communication-type paper.

Round 2

Reviewer 2 Report

The effort to address the critique is appreciated, however, there are still problems to be addressed.

The authors reply has convinced me that they are not fully familiar with principle regulating melanogenesis.

Melanocytes (epidermal human): this will require  future experiments; please discuss this weakness in limitations. Again B16 is not representative model it is  murine melanoma, which may have limited application to human system.

DOPA oxidase, please provide the concentration of the substrate in mM and state whether DOPA solution in buffer was used as the blank. Note, DOPA undergoes an  autooxidation in a process dependent on pH and metal cations. Tyrosine hydroxylase is the rate limiting reaction. People use dopa oxidase because it is easy to assay and tyrosine hydroxylase requires some expertise in biochemistry.

Use of DMEM media that contains high concentration of tyrosine can affect cellular phenotype (J Cell Sci 89, 287-296, 1988; Pigment Cell Res 2. 109-116, 1989). The authors evaded or did not understand my previous question. This issue should be briefly discussed.

Schematic figure show pathways regulating melanogenesis  (applicable in murine system) without specifying where that test compounds act. In fact it is disconnected from the results. Better not to show it, but briefly mention proposed mechanism of action. Unless the authors want to run additional mechanism  oriented experiments to provide the proof for the proposed model.

Author Response

Thank you for your useful comments and suggestions on the structure of our manuscript. We have modified the manuscript accordingly, and detailed corrections are listed below point by point. We will be making corrections in English for the newly changed and inserted content.

[Reviewer 2]

The effort to address the critique is appreciated, however, there are still problems to be addressed. The authors reply has convinced me that they are not fully familiar with principle regulating melanogenesis.

(1) Melanocytes (epidermal human): this will require  future experiments; please discuss this weakness in limitations. Again B16 is not representative model it is  murine melanoma, which may have limited application to human system.

→ We have corrected the limitations for B16F10 cells that you pointed out as follows (L285-292). We will be making corrections in English for the newly changed and inserted content.

“However, our study had several limitations. The mouse melanoma cancer cell B16F10 is widely used as an experimental cell line in many studies on melanogenesis. Therefore, we used the same cell line for experiments in this study. Although bergapten showed the melanogenic effect at the cellular level, these results do not always provide the same outcomes as human melanocytes or clinical manner. Therefore, for practical application of the results, further human melanocytes or clinical studies will be required to determine the therapeutic regimen of loratadine for treatment of hypopigmentary disorders in humans.”

(2) DOPA oxidase, please provide the concentration of the substrate in mM and state whether DOPA solution in buffer was used as the blank. Note, DOPA undergoes an  autooxidation in a process dependent on pH and metal cations. Tyrosine hydroxylase is the rate limiting reaction. People use dopa oxidase because it is easy to assay and tyrosine hydroxylase requires some expertise in biochemistry.

→ We have revised the DOPA concentration and the blank you requested (L346-348). We will be making corrections in English for the newly changed and inserted content.

(3) Use of DMEM media that contains high concentration of tyrosine can affect cellular phenotype (J Cell Sci 89, 287-296, 1988; Pigment Cell Res 2. 109-116, 1989). The authors evaded or did not understand my previous question. This issue should be briefly discussed.

→ We read the paper you introduced in depth and inserted the following contents into materials and methods (L324-329). We will be making corrections in English for the newly changed and inserted content.

“L-tyrosine and L-dopa, two major substrates of the melanogenesis pathway, have a positive regulation of melanin pigmentation. Many studies reported that B16 mouse melanoma cells were cultured in DMEM characterizing high concentration of tyrosine (0.4 mM) that can affect cellular phenotype and melanogenesis [47-49]. Therefore, we conducted all experiments within 84 h of the start of the log phase of B16F10 cells in order to minimize the unexpected role of DMEM medium.”

(4) Schematic figure show pathways regulating melanogenesis  (applicable in murine system) without specifying where that test compounds act. In fact it is disconnected from the results. Better not to show it, but briefly mention proposed mechanism of action. Unless the authors want to run additional mechanism oriented experiments to provide the proof for the proposed model.

→ We deleted the schematic figure.